# Broadband Antireflective Hybrid Micro/Nanostructure on Zinc Sulfide Fabricated by Optimal Bessel Femtosecond Laser

**DOI:** 10.3390/nano13071225

**Published:** 2023-03-30

**Authors:** Xun Li, Ming Li

**Affiliations:** State Key Laboratory of Transient Optics and Photonics, Xi’an Institute of Optics and Precision Mechanics of CAS, Xi’an 710119, China

**Keywords:** femtosecond laser, infrared antireflection, micro/nano hybrid structure

## Abstract

Enhancing the infrared window transmittance of zinc sulfide (ZnS) is important to improve the performance of infrared detector systems. In this work, a new hybrid micro/nanostructure was fabricated by an optimal Bessel femtosecond laser on ZnS substrate. The surface morphologies and profiles of ASS ablated by a 20× microscope objective Bessel beam are described, indicating that the nanoripples on the micropore were formed by the SPP interference and the SPP scattering in a particular direction. Further, the maximum average transmittance of ASS increased by 9.7% and 12.3% in the wavelength ranges of 5~12 μm and 8~12 μm, respectively. Finally, the antireflective mechanism of the hybrid micro/nanostructure is explored using the novel electromagnetic field model based on the FDTD method, and we attribute the stable antireflective performance of ASS in broadband to the interface effective dielectric effect and LLFE.

## 1. Introduction

In consideration of its stable chemical and physical properties in harsh and thermal shock environments, zinc sulfide (ZnS) can maintain more than 50% transmittance in the infrared band of 4~12 μm [1,2,3], making it a suitable infrared window material for military infrared detection systems [4], including aircraft optoelectronic tracking systems, laser guidance, and night vision monitoring. However, the high refractive index (*n* = 2.2 at 10.6 µm) of bare ZnS leads to more surface reflection losses (about 25%), affecting the quality of infrared photoelectric systems [5,6]. The conventional method to realize antireflection is coating single-layer or multilayer films on the surface, inevitably leading to drawbacks such as thermal stress mismatch, narrow bandwidths, polarization sensitivity, and limited acceptance angle [7,8,9]. More seriously, film shedding or cracking is frequently encountered owing to the thermal and lattice mismatches between the coatings and the substrate [10,11]. An approach that has proven effective in reducing Fresnel reflections while also mitigating the issues associated with traditional coatings is the direct fabrication of an antireflective subwavelength structure (ASS) in the surfaces of optics [12,13,14]. The existing ASS manufacturing approaches, including machining [15], reactive ion etching [16], chemical etching [17], electron beam lithography [18], lithography [19], and imprinting [20], have some limitations, such as negative environmental impacts, limited materials, high costs, multiple steps, and time-consuming and difficult fabrication of curved parts.

Due to its distinct advantages such as one-step shaping, flexible machining of space, high processing accuracy, and being maskless and environmentally friendly [21,22], the femtosecond laser is a promising technology for fabricating ASS on various materials in one step [23,24,25,26,27]. Tarabrin et al. fabricated an antireflective microstructure by a monopulse femtosecond laser on AgClBr fiber, which increased the transmittance of the fiber to 92.8% at a 10.6 µm wavelength [28]. Bushunov et al. adopted several methods to fabricate ASS on chalcogenide crystals, including direct single-pulse ablation, in-depth focusing, additional spherical aberration, and obstruction of peripheral rays, and finally achieved the transmittance of 97% in the 2.7–8 µm wavelength range [29]. Wang et al. proposed a double-pulse femtosecond laser to fabricate a quasi-periodic array on ZnS substrate, and observed an increase in average transmittance of 7.6% at a 8.1 μm wavelength [30]. Zhang et al. designed and fabricated an antireflective structure by a femtosecond laser-shaped beam on ZnS, resulting in transmittance enhancement in the 5.5–14 μm wavelength range, expressing broadband response and a large acceptance angle [31]. Several research teams [4,32,33] have agreed that the depth of ASS is a key factor to improve the antireflective performance, since it determines the smoothness of an effective refractive index gradient. However, the depth of ASS processed by the Gaussian beam with a small focal depth and concentrated energy is insufficient, which suppress the transmittance within the infrared band [34].

In this study, an optimal Bessel femtosecond laser is utilized to prepare hybrid micro/nanostructures on ZnS substrate for suppressing surface Fresnel reflection. The surface morphology is regulated by a 20× microscope objective Bessel beam, and the evolution of nanoripples on hybrid micro/nanostructures is discussed. Furthermore, the profile and the antireflection performance of ASS with three different periods are characterized. Finally, the optical field intensity distributions around ASS are simulated by the finite difference time domain (FDTD) method, which is used to discuss the broadband antireflective mechanism of the hybrid micro/nanostructure.

## 2. Materials and Methods

In this experiment, a linearly polarized femtosecond laser (Pharos, Light Conversion, Suwalki, Poland) with an operating wavelength of 1030 nm, a repetition rate of 1000 kHz, and a pulse width of 290 fs was adopted to generate the Gaussian beam. The original output beam with a diameter of 10.8 mm was shaped into a Bessel beam by using an axicon lens with a taper angle of 2° (AX252-B, Thorlabs, Newton, NJ, USA). In order to obtain a micrometer-scale ASS on the substrate, a convex lens (focal length = 155 mm) and a 20× objective lens (focal length = 10 mm, NA = 0.42) (NIR-20-95, Tokyo, Japan) were utilized to further compress the Bessel beam focal depth into a length of 960 μm and the beam focus spot into a diameter of 3.2 μm.

A ZnS sample with a size of ϕ25.4 × 2 mm was polished to an optical grade without coating and installed on a three-dimensional high-precision motion platform (P-622.2CD, PI, Auburn, MA, USA) with a movement repeatability of 2 nm. Then, the sample was moved along the X/Y axis by a distance of L, which is the period of the microstructure array. By adjusting the laser pulse energy and the number of pulses, the periodic arrangement of moth-eye microstructures was prepared. Subsequently, the surface morphologies of the samples were characterized by a scanning electron microscope (SEM, SU8010, Hitachi, Tokyo, Japan). The profiles of the structures were observed by atomic force microscopy (AFM, Innova, Bruker, Leipzig, Germany). Ultimately, the infrared transmittance of the ZnS was measured by an infrared microscope (FTIR, Bruker, vertex70+Hyperion1000, Leipzig, Germany).

In the simulation, the optimal ASS profile inspired by moth-eye structures was modeled on the ZnS substrate by the finite difference time domain (FDTD) method [35] to explore its antireflective mechanism. The 3D model of the hybrid micro/nanostructure, which consists of micropores and nanoripples, was established with solid works and imported into Lumerical in STL format for anti-reflection simulation. The transmission line treatment and the perfectly matched layer (PML) boundary condition were employed in the direction of the incident beam propagation (Z direction) and the other two vertical directions of beam propagation (X and Y directions), respectively. A block/periodic plane wave with a wavelength range of 5–12 μm was used as an infrared light source aiming to simulate the infrared measurement. The mesh size of the simulated grid was set to 5 nm to improve the accuracy of results. The simulated time was set to 1000 fs to obtain stable results of the electric field component (Ex) distribution, transmittance, and reflectance of the designed ASS.

## 3. Results and Discussion

Zero-order Bessel beams have a decisive advantage over Gaussian beams [36] in creating deeper micropores [33]. In particular, explained from the characteristics of penetration geometry and nonlinear robustness, there are obvious differences in ASS on the ZnS surface fabricated by a Bessel beam with various light field distributions [37,38]. Hence, it is necessary to optimize the Bessel beam to be more suitable for the fabrication of ASS, so as to improve the depth and quality of the structure. Figure 1 illustrates the surface morphologies of structures on ZnS fabricated by different Bessel beam parameters, with the pulse energy and pulse number varying from 4 µJ to 2 µJ and from 1 × 10^6^ to 1 × 10^4^, respectively. The laser fluence of the zero-order and the first-order diffraction rings of the Bessel beam was 9.9 J/cm^2^ and 0.39 J/cm^2^ with a pulse energy of 1.8 μJ, respectively. Since the accumulated energy of multiple pulses was still higher than the ablation threshold of 0.48 J/cm^2^ [33], the molten material in the cavity was sprayed, adhered to, deposited, and then cooled to the edge of the hole, forming the recast layer during the laser ablation, as depicted in Figure 1a–c. The morphology of the microstructure gradually evolved into a clear micropore as the pulse number decreased from 1 × 10^6^ to 1 × 10^4^ for the pulse energy of 2 µJ, while the volcanic structure was still formed by the molten material sticking to the edge of the hole, as demonstrated in Figure 1d–f. Subsequently, the laser fluence of the zero-order diffraction ring decreased to 4.9 J/cm^2^ with a pulse energy of 2 µJ, and a clean and flat single hole without sidelobes was obtained by setting the laser energy near the laser ablation threshold, as exhibited in Figure 1g–i.

After laser irradiation, the laser-induced period surface structures (LIPSSs) with the period of 200–300 nm emerged in the micropores. Due to its period being less than λ/2, the periodic structure is defined as High Spatial Frequency LIPSS (HSFL), the orientation of which is always perpendicular to the laser polarization simultaneously. This phenomenon originated from the excitation of surface plasmon polaritons (SPPs) and the interference between surface plasmons and the incident laser. The Sipe–Drude model expresses this process; the interference of the electric field of the SPPs with the incident laser leads to a spatially modulated energy deposition into the material and finally generates the LSFL via inhomogeneous ablation [39,40,41]. In addition, the initial surface ripple formed by the previous pulse causes the SPP to scatter vertically along the direction of the nanoripple, which enhances the surface plasma wave conversion and further deepens the nanoripples [42,43,44].

The period of ASS can be deduced from a grating diffraction equation for inhibiting nonzero diffraction [33], and then ASS was fabricated with three periods of 2.6 μm, 3 μm, and 3.6 μm, as exhibited in Figure 2. The surface morphology of the micro/nanostructure was fabricated by the Bessel beam with single pulse energy of 2 μJ and a pulse number of 1 × 10^5^. Obviously, the areas between the moth-eye-type array were still smooth without splashes or cracks, indicating that the cumulative thermal effects were suppressed effectively.

Figure 3 shows the three-dimensional profiles of the subwavelength quasi-periodic array (SQA) characterized by atomic force microscopy. The depth of SQA ablated by a 20× microscope objective Bessel beam was about 1 μm, which is twice as deep as fabrication by a 50× microscope objective Bessel beam [33]. Meanwhile, the surface of SQA measured in a square region of 20 × 20 µm was relatively smooth with surface roughness from 26.3 nm to 30.6 nm. The ultra-low roughness can reduce light scattering and improve light transmittances of SQA, which is primarily benefited by the small heat-affected zone during ultrafast laser ablation.

Additionally, Figure 2 and Figure 3 demonstrate that the 20× microscope objective Bessel beam had a lower thermal-affected zone than the 50× microscope Bessel beam, and it increased the depth of ASS from 0.5 μm to 1 μm [33]. This is attributed to the focal depth of the 20× microscope objective Bessel beam being 1.5 times greater than that of the 50× microscope objective Bessel beam, which facilitates the fabrication of deeper micropores in transparent materials from the perspective of penetrating geometric optics. Moreover, the laser fluence of the zero-order diffraction ring of the 20× microscope Bessel beam is 10 times less than that of the 50× microscope Bessel beam [33]. Such lower light intensity is conducive to conquering undesirably nonlinear effects of the laser beam before it converges, including Kerr self-focusing, multiphone ionization, avalanche ionization, plasma volume shielding, defocusing, etc. Thus, optimizing the intensity distribution of the Bessel beam is essential to improve the depth and quality of ASS structure for different substrate materials.

Figure 4a shows that the transmittances of the fabricated ASS increase gradually with the period decreasing from 3.6 μm to 2.6 μm. The ASS prepared by the 20× objective Bessel beam achieved broadband antireflection in the infrared band [33; the transmittance of ASS with three periods increased by 7.1% to 9.7% in the 5–12 µm wavelength range and obtained the maximum value of 9.7% for the period of 2.6 μm. The maximum average transmittance of ASS increased by 12.3% in the wavelength range of 8 μm~12 μm. This illustrates that a small heat-affected zone facilitates smaller periodic array fabrication, which weakens the diffraction and scattering effects of ASS in the mid-infrared region [45]. Additionally, the equivalent refractive index gradient of ASS becomes more linear and gentler with the increase in the array depth, and effectively reduces light interference between the entrance and the bottom of ASS. Figure 4b shows that the reflectance of ASS with three different periods decreased by 3.8%, 2.9%, and 1.8%. The ASS reduced the reflectance of the ZnS surface and part of the energy was converted into transmitted light while the other part was absorbed, including resonance absorption [46], nanoripple absorption [47], and nanoparticle scattering. Apparently, the nano periodic ripples inside the microstructure played an important role.

In order to explore the antireflective mechanism of the hybrid micro/nanostructure, the electric field (light field) intensity distributions for ASS with various periods of ripples were calculated in the wavelength range of 5 μm–12 μm. The transmission spectra fluctuated gently, and the reflectance spectra oscillation basically disappeared, as depicted in Figure 5. Additionally, the transmittance of ASS with various numbers of ripples showed a slight improvement compared with a single parabolic hole, and the maximum average transmittance increased by 0.4%. This is attributed to the dielectric environment of the ZnS interface altered by nanoripples, which is conducive to reducing the light reflection on the wall of the micropore. Furthermore, the nanoripples emerge the action of local light field enhancement (LLFE) [31], which can effectively capture incident photons and eliminate surface Fresnel reflection. Apparently, compared with improving the spectral transmittance of ASS, the hybrid micro/nanostructure is mainly conducive to reducing the transmittance fluctuation in a wider spectrum. Compared with the electric field intensity of a single parabolic micropore in Figure 6a,b, the infrared light intensity is mostly immersed in the hybrid micro/nanostructure, proving LLFE in the whole wavelength, as shown in Figure 6c–f. It is proved that the hybrid micro/nanostructure with nanoripples exhibits better antireflection in the ultra-wide wavelength.

Figure 7a shows that compared with a single parabolic hole, the average transmittance of SQA increases from 0.1% to 0.36% as the height of ripples decreases from 400 nm to 100 nm. Figure 8 shows the optical field intensity distributions of the SQA with four ripples at the central wavelength of 8 μm. Obviously, the LLFE of SQA still compresses incident photons to the bottom of the micropore by the nanoripples, inhibiting the interference intensity of the light field at the entrance of the SQA. Figure 7b shows that the reflectance of nanoripples first decreases by 0.2% at the height of 100 nm, and then increases from 0.04% to 0.07%, with the ripples’ height increasing from 200 nm to 400 nm. This is because the height of the nanoripples gradually increases to the same size of the parabolic micropores, resulting in the surface effective dielectric effect and LLFE becoming gradually ineffective. Consequently, the antireflective performance of the hybrid micro/nanostructure is the balance between the subwavelength structure and the nanostructure.

## 4. Conclusions

In summary, a new hybrid micro/nanostructure was fabricated by an optimal Bessel femtosecond laser on ZnS substrate, and the geometrical morphology evolution and the antireflective mechanism of the hybrid micro/nanostructure were explored. Initially, a 20× microscope objective Bessel beam was designed and optimized. Then, the surface profiles after the multi-pulse Bessel beam ablation were revealed, and the nanoripples on the micropore were formed by the interference between the SPP and the incident laser, as well as the SPP scattering in a particular direction. Moreover, ASS with three different periods was fabricated by the optimal femtosecond laser Bessel beam, and the average transmittance of ZnS increased by 9.7% in the wavelength range of 5 μm~12 μm for the period of 2.6 μm, which indicated that the hybrid micro/nanostructure obtained high transmittance over a wide bandwidth. Eventually, the novel electric field intensity distributions around ASS were simulated by the FDTD method, and the hybrid micro/nanostructure demonstrated more stable antireflective performance in broadband, attributed to the interface effective dielectric effect and LLFE.

## Figures and Tables

**Figure 1 nanomaterials-13-01225-f001:**
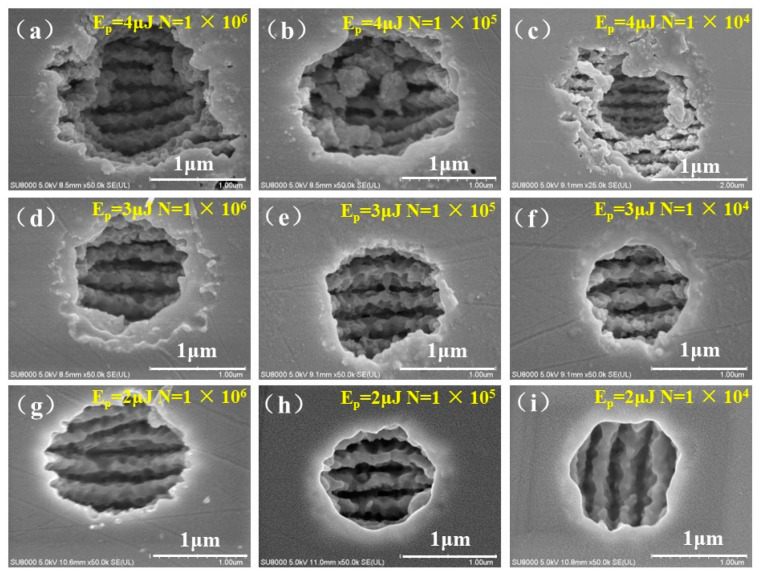
The surface morphologies of micropores ablated by Bessel beam with different pulse energies and pulse numbers. (**a**) pulse energy and pulse number were 4 µJ and 1 × 10^6^, respectively; (**b**) pulse energy and pulse number were 4 µJ and 1 × 10^5^, respectively; (**c**) pulse energy and pulse number were 4 µJ and 1 × 10^4^, respectively; (**d**) pulse energy and pulse number were 3 µJ and 1 × 10^6^, respectively;(**e**) pulse energy and pulse number were 3 µJ and 1 × 10^5^, respectively; (**f**) pulse energy and pulse number were 3 µJ and 1 × 10^4^, respectively; (**g**) pulse energy and pulse number were 2 µJ and 1 × 10^6^, respectively; (**h**) pulse energy and pulse number were 2 µJ and 1 × 10^5^, respectively; (**i**) pulse energy and pulse number were 2 µJ and 1 × 10^4^, respectively.

**Figure 2 nanomaterials-13-01225-f002:**
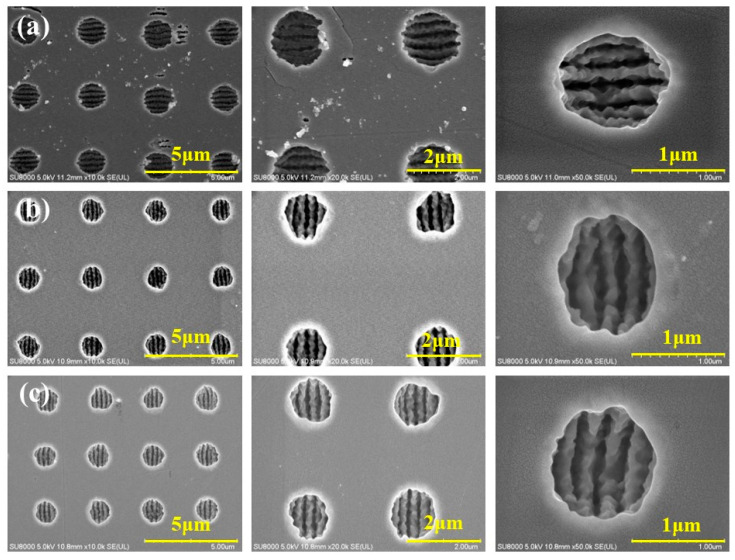
SEM images of micropore arrays with different periods. (**a**) 3.6 µm period; (**b**) 3.0 µm period; (**c**) 2.6 µm period.

**Figure 3 nanomaterials-13-01225-f003:**
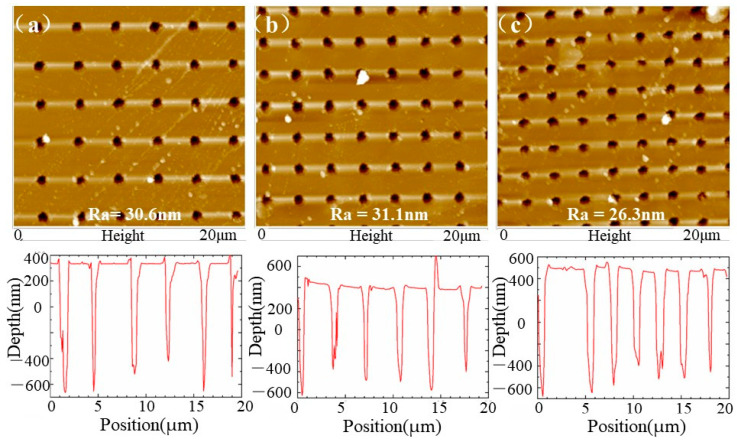
AFM images of micropore arrays with different periods. (**a**) 3.6 µm period; (**b**) 3.0 µm period; (**c**) 2.6 µm period.

**Figure 4 nanomaterials-13-01225-f004:**
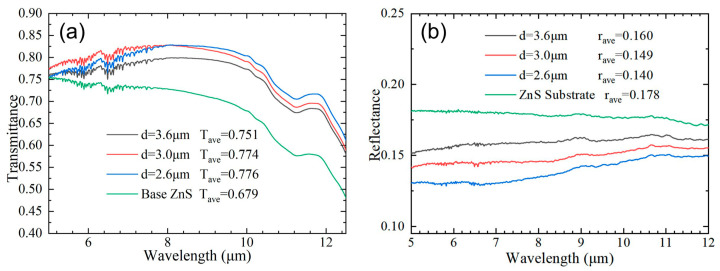
Measured infrared transmittance (**a**) and reflectance (**b**) spectra of ASS fabricated by 20× microscope objective Bessel beam.

**Figure 5 nanomaterials-13-01225-f005:**
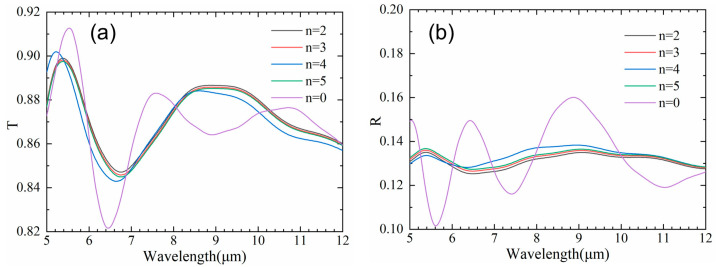
Simulated infrared transmittance (**a**) and reflectance (**b**) spectra of hybrid micro/nanostructure with various numbers of nanoripples.

**Figure 6 nanomaterials-13-01225-f006:**
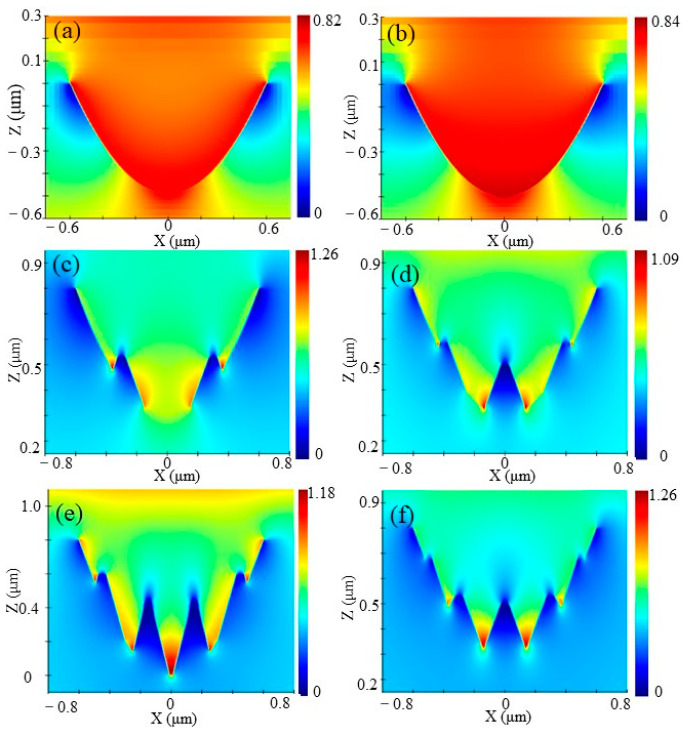
Optical field distribution of parabolic holes in the incident wavelength of 8 μm (**a**) and 12 μm (**b**), and nanoripple numbers of 2 (**c**), 3 (**d**), 4 (**e**), and 5 (**f**) in the incident wavelength of 8 µm.

**Figure 7 nanomaterials-13-01225-f007:**
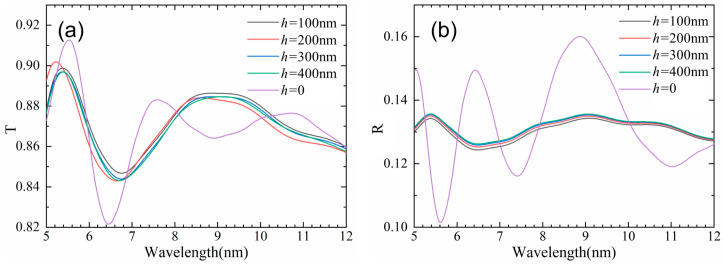
Simulated infrared transmittance (**a**) and reflectance (**b**) spectra of hybrid micro/nanostructure with different nanoripple heights.

**Figure 8 nanomaterials-13-01225-f008:**
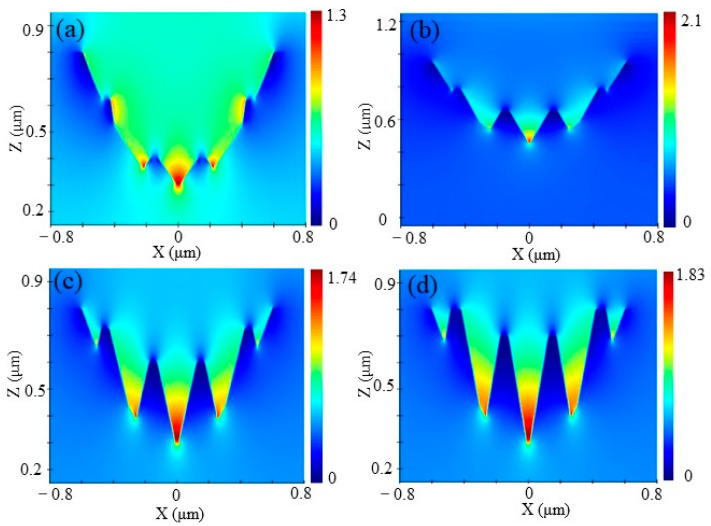
Light field distribution of the hybrid micro/nanostructure with heights of 100 nm (**a**), 200 nm (**b**), 300 nm (**c**), and 400 nm (**d**).

## Data Availability

The study did not report any data.

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
