# Peer review of "Broadband Antireflective Hybrid Micro/Nanostructure on Zinc Sulfide Fabricated by Optimal Bessel Femtosecond Laser"

_nanomaterials, 2023, doi:10.3390/nano13071225_

Round 1

Reviewer 1 Report

Article Ref.

nanomaterials-2229226

Title:

Fabrication of Micro/nano Hybrid Structure for Broadband Antireflection by Multi-pulse Bessel Femtosecond Laser Direct writing

Comments to the authors:

This work reports the fabrication of a sort of hierarchical micro/nanostructure comprised of micron sized holes with a nanograting like structure inside. The authors employ a Bessel femtosecond laser for the direct patterning of a ZnS substrate and characterize the influence of the structure in the reflectance and transmittance of the films in the MIR region. Also, the authors include some simulations to evaluate the effect of the nanostructure morphology on the antireflective properties of the material.

 The topic of the work is of interest, and the fabrication shown by the authors seems to be promising, specially if it were possible to achieve a broader portfolio of the hierarchies generated, for instance by using circularly polarized beams.

 However, from the point of view of this reviewer, the paper lacks clarity, with some confusing paragraphs, and some discrepancies throughout the text that need to be explained. To point out some of the main ones:

-        The authors state that “The SQA depth from 20× microscope objective Bessel beam irradiation is almost 1 μm, which is twice as much as that from 50× microscope objective Bessel beam irradiation with a value of 0.5 μm”. The depth of the holes shown in the depth profiles in Figure 3 (which actually is not mentioned in either the text or the caption figure) indicates maximum values of 700-800 nm and a near mean value that can be estimated close to 500-600, far from that claimed.

-        The explanation for the formation of the periodic ripples is not clear and it is not supported either by experimental results or other references. The pattern remains to that observed in LIPSS with the difference in grating orientation respect to the polarization direction.

-        The simulations included in the work does not agree with the experimental results. Even the transmittance and reflectance curves of the smooth ZnS samples are quite different quantitatively and qualitatively.

-        Figure 4. It is expected that the microstructure providing for the higher transmittance shows the lower reflectance, but there is a reversal in order comparing the microstructures with d=3 and d=2.6.

-        The simulations presented show a minor influence of both, the number and height of the ripples in the transmittance in the IR region studied, contrary to what is claimed in the text, with variations as low as 0.04%. The differences would probably be more pronounced in the visible region than in the IR region studied, as has been demonstrated in some of the references included in the work. 

From the point of this reviewer this work cannot be accepted for publication and requires a profound revision to be submitted again to Nanomaterials.

Author Response

Review1: The authors state that “The SQA depth from 20× microscope objective Bessel beam irradiation is almost 1 μm, which is twice as much as that from 50× microscope objective Bessel beam irradiation with a value of 0.5 μm”. The depth of the holes shown in the depth profiles in Figure 3 (which actually is not mentioned in either the text or the caption figure) indicates maximum values of 700-800 nm and a near mean value that can be estimated close to 500-600, far from that claimed.

Response1: The depth of micro/nano structure characterized by the AFM will be different due to different measured position of ZnS. According to the reviewer's opinion, the measurement position is re-selected, indicating that the depth of ASS ranges from 800μm to 1100μm,as shown in Fig. 3.

Review2: The explanation for the formation of the periodic ripples is not clear and it is not supported either by experimental results or other references. The pattern remains to that observed in LIPSS with the difference in grating orientation respect to the polarization direction.
Response2: According to the reviewer's opinion, the formation of the periodic ripples is explained (lines 118-129), and relevant references are attached to support my opinion, as exhibited in reference [37] to reference [ 42].

Review3: The simulations included in the work does not agree with the experimental results. Even the transmittance and reflectance curves of the smooth ZnS samples are quite different quantitatively and qualitatively.

Response3:About“The simulations included in the work does not agree with the experimental results”. It is a common phenomenon that the experimental results differ from the simulation results [3,4]. The surface scattering from the recast layer and the surface undulation in fabricated sample are difficult to simulate in FDTD.  Therefore, the deviations of experimental and simulation due to light interference between upper and lower ideal smooth surface of ZnS. [7,8]. Additionally, Nanoripples on hybrid micro-nano structure is difficult to be accurately measured, so nanoripples can only be simulated according to the theoretical morphology. The above two factors lead to the simulations does not agree with the experimental results.

 As for the query that the “the transmittance and reflectance curves of the smooth ZnS samples are quite different quantitatively and qualitatively”. First of all, some literatures reported that the transmittance of ZnS is 68% to 75% in the wavelength of 2.5μm-12μm [1,2]. The difference of transmittance is mainly due to the various purity and thickness of zinc sulfide prepared by chemical vapor deposition (CVD). Secondly, in many published results [3-6], it is common that the spectral curves of smooth ZnS samples are differ quantitatively and qualitatively. Finally, the actual transmittance and reflectance measured in this experiment are the average values in the range of 5μm~12μm, and it is normal to have differences with the spectral transmittance in a specific wavelength (n = 2.2 at 10.6 µm) (line 26).

Review4: Figure 4. It is expected that the microstructure providing for the higher transmittance shows the lower reflectance, but there is a reversal in order comparing the microstructures with d=3 and d=2.6.

Response4: The microstructure with the period of d=2.6μm actually obtained the lowest reflectance, The reason leading to the reviewer's misjudgment are the color of transmission spectra inconsistent with reflection spectra.  Therefore, the colors of transmission spectra and reflection spectra have been revised,as exhibited in FIG.4.

Review5: The simulations presented show a minor influence of both, the number and height of the ripples in the transmittance in the IR region studied, contrary to what is claimed in the text, with variations as low as 0.04%. The differences would probably be more pronounced in the visible region than in the IR region studied, as has been demonstrated in some of the references included in the work.

Response5: As stated by the reviewer, the transmittance of the ASS with different number of ripples are all silightly improved compared with single parabolic hole (rather than compare with ZnS substrate), and the maximum average transmittance increases by 0.4% (rather than increases to 0.4%). Moreover, compared with improving the transmittance of ZnS, such hybrid micro/nano structure is mainly conducive to reducing the transmittance and reflectance fluctuation in a wider spectrum. The reviewer's misunderstanding may be caused by the author's unclear elaboration, so the author re-writes this section (line 186-202).

Reviewer 2 Report

In this work, the authors introduce a comprehensive study on the fabrication of high-transmittance ZnS infrared windows by femtosecond laser micro/nanostructuring.

I do not discuss about the scientific quality of the work, which seems to be good. Experimental results are interesting, thoroughly discussed, and validated by FDTD simulations. Methodology is carefully and accurately described. The usefulness of the work is undoubtedly high, being far-infrared windows a crucial component for defense application.

By the way, I find that the paper should be deeply revised for several reasons:

1) English should be definitely improved. Some paragraphs are unreadable, using expressions such as "help conquered" (line 185, but it's just an example) which are really hard to understand. Verbs, adjectives, are very often wrong. It was very hard for me to follow the thread of the speech in some points, really. I strongly recommend the authors to improve English.

2) Being ZnS structures, I'm surprised that "ZnS" or "Zinc Sulphide" are not present in the title of the paper. Authors should specify in the title the material subjected to femtosecond laser treatment.

3) Even if the introduction effectively brings the reader deep into the topic of the paper, the novelty of the work and of the presented results is not clear to me, and should be better and clearly highlighted by the authors, for instance with respect to the paper “Antireflective array on zinc sulfide fabricated by femtosecond laser” Optics & Laser Technology 155  (2022) 108438, which looks very similar to the submitted manuscript.

4) Finally, authors dedicate a whole paragraph (lines 127-141) discussing on the "periodic nanogratings" appearing on (and not "in") the wall of the microhole, and on their origin. For this purpose, they cite a paper [Ref. 37], published 50 years ago (1973). Probably the authors missed the fact that the formation of periodic nanograting by femtosecond laser treatment when the laser fluence is above the ablation threshold is a well-known phenomenon. More specifically, in their case, such nanogratings are high-spatial-frequency LIPSS (Laser-Induced Periodic Surface Structures), the period of which is about lambda/2n, where "lambda" is the laser wavelength and n is the refraction index of the material. Moreover, it's not true that nanogratings are obtained only in the case of Bessel beams: this is true also in the case of Gaussian beams. There are plenty of papers published in recent years focused on LIPSS. I suggest the authors to re-write the section by considering the scientific advancements in the topic.  Some references on LIPSS: 1) Materials 2022, 15(4), 1378; 2) Journal of Laser Applications 24, 042006 (2012); 3) Nanomaterials 2020, 10(10), 1950.

Minor comments:

-       Line 30: provide some examples of materials used for antireflection films

-       Line 36: it’s not “reactive ion lithography” but “reactive ion etching”.

-       It would be better to replace “energy density” with “laser flunce”. It’s more correct.

-       Line 131: spell “SPP” acronym.

-       Figure 1: please shorten the caption! The sentence “The microholes were ablated…etc.” is repeated nine times! I would insert in the figure the pulse energy and number values, instead, without repeating every time the same thing.

-       Line 134: spell “SWMS” acronym.

-       Line 165: spell “SQA” acronym.

-       Line 177: authors refer to “Figure 5” showing light field distribution characteristics at 20x and 50x. But I do not understand. Actual Figure 5 shows simulated T and R spectra. Probably, author mean Figure 5 of Reference [31], which they cite to make a comparison between 20x and 50x. If so, the authors should specify it.

-       Line 179: I think there’s something wrong with the value 39.4 J/cm2, I think it’s higher: it cannot be equal to the first-order value mentioned at line 180. Please check.

-       Line 203: replace “Bezier” with “Bessel”.

-       Line 209: authors should mention that increased IR absorbance by fs-laser induced nanogratings is well-known result from the literature (Appl. Phys. A 122, 211 (2016)).

Author Response

-Review1: English should be definitely improved. Some paragraphs are unreadable, using expressions such as "help conquered" (line 185, but it's just an example) which are really hard to understand. Verbs, adjectives, are very often wrong. It was very hard for me to follow the thread of the speech in some points, really. I strongly recommend the authors to improve English.

Response1: I have revised the whole manuscript according to the reviewer's opinion.

-Review2: Being ZnS structures, I'm surprised that "ZnS" or "Zinc Sulphide" are not present in the title of the paper. Authors should specify in the title the material subjected to femtosecond laser treatment.

Response2: I have revised the manuscript according to the reviewer's opinion, the title have revised to“Broadband Antireflective Hybrid Micro/nano Structure on Zinc Sulfide Fabricated by Optimal Bessel Femtosecond Laser”.

-Review3: Even if the introduction effectively brings the reader deep into the topic of the paper, the novelty of the work and of the presented results is not clear to me, and should be better and clearly highlighted by the authors, for instance with respect to the paper “Antireflective array on zinc sulfide fabricated by femtosecond laser” Optics & Laser Technology 155 (2022) 108438, which looks very similar to the submitted manuscript.

Response3: The novelty of the work is better and more clearly emphasized, as shown in manuscript. 1. The Bessel beam has been optimized, in order to be more suitable for processing of micro/nano structure on zinc sulfide. 2. Although many studies have pointed out that the antireflection structure belongs to micro/nano structure, nanostructures are distributed on the surface of micropores. But, the SEM images of many references indicate that the structure of this study is different from that of other references [1-5]. The distribution of the nanostructure in this research is more regular, which belong to nanoripples. Meanwhile, the period of nanoripples is also larger. 3.The novel electric field intensity distributions around hybrid micro/nano structure are simulated by the FDTD method, so as to explore the antireflective mechanism of the micro/nano hybrid structure.

The difference between the manuscript and the paper (Optics & laser Technology 155(2022) 108438) is that:1. The method and experiment for forming the Bessel beam is different: in the reference, an axicon lens with a taper angle of 175â—¦ was used to form a Bessel beam, and then focused by a lens with a focal length of 50 mm. In this manuscript, an axicon lens with a taper angle of 2â—¦ was adopted, and then, a convex lens (focal length = 155 mm) and a 20 × objective lens (focal length = 10 mm,NA = 0.42) were utilized to further compress.2. The light field distribution of Bessel beam is different. Attributing to only a lens with a focal length of 50 mm was used in the reference, so the diameter of the focal spot was approximately 30 μm. While the Bessel beam focal depth into length of 960 μm and beam focus spot into diameter 3.2μm in this manuscript.3. Micro/nano structure morphology is different: the reference stated that its micro/nano structure was micropore, while a micro/nano hybrid structure with nanoripples was formed in the manuscript. 4. Antireflection bandwidth is different. the wavelength of ASS for antireflection is from the ultraviolet region to the near-infrared region in the reference, while the wavelength range of the ASS in this manuscript is from 5 μm to 12 μm. 5. The research emphases of the articles are different: The main study in the reference are the wide-angle antireflection (light absorption) and hydrophobicity of the ASS, while this manuscript focuses on the antireflection (light transmittance) properties and the anti-reflection mechanism of micro/nano structures.

-Review4: Finally, authors dedicate a whole paragraph (lines 127-141) discussing on the "periodic nanogratings" appearing on (and not "in") the wall of the microhole, and on their origin. For this purpose, they cite a paper [Ref. 37], published 50 years ago (1973).  Probably the authors missed the fact that the formation of periodic nanograting by femtosecond laser treatment when the laser fluence is above the ablation threshold is a well-known phenomenon.  More specifically, in their case, such nanogratings are high-spatial-frequency LIPSS (Laser-Induced Periodic Surface Structures), the period of which is about lambda/2n, where "lambda" is the laser wavelength and n is the refraction index of the material.  Moreover, it's not true that nanogratings are obtained only in the case of Bessel beams: this is true also in the case of Gaussian beams.  There are plenty of papers published in recent years focused on LIPSS. I suggest the authors to re-write the section by considering the scientific advancements in the topic. Some references on LIPSS: 1) Materials 2022, 15(4), 1378; 2) Journal of Laser Applications 24, 042006 (2012); 3) Nanomaterials 2020, 10(10), 1950.

Response4: I have revised the manuscript according to the reviewer's opinion (line 118-129). Meanwhile, six new references have been cited, including three references recommended by the reviewer(reference40-45).

Minor comments:

-Line 30: provide some examples of materials used for antireflection films

-Response1: three references have been cited (line 30, reference 7 -9)

-Line 36: it’s not “reactive ion lithography” but “reactive ion etching”.

-Response2: The relevant content has been corrected according to the reviewer's suggestion (line 35- 36).

-It would be better to replace “energy density” with “laser flunce”.  It’s more correct.

-Response3: The relevant content has been corrected according to the reviewer's suggestion.

- Line 131: spell “SPP” acronym.

-Response4: The relevant content has been revised according to the reviewer's suggestion (line 122).

-Figure 1: please shorten the caption!  The sentence “The microholes were ablated…etc.” is repeated nine times!  I would insert in the figure the pulse energy and number values, instead, without repeating every time the same thing.

-Response5: The figure has been revised according to the reviewer's suggestion, as illustrated in Fig 1.

- Line 134: spell “SWMS” acronym.

-Response6: The “SWMS” has been deleted.

- Line 165: spell “SQA” acronym.

- Response7: The relevant content has been revised according to the reviewer's suggestion (line 142).

-Line 177: authors refer to “Figure 5” showing light field distribution characteristics at 20x and 50x. But I do not understand. Actual Figure 5 shows simulated T and R spectra.  Probably, author mean Figure 5 of Reference [31], which they cite to make a comparison between 20x and 50x.  If so, the authors should specify it.

-Response8: This part is really not clear, so this section has been revised (line 152-167).

-Line 179: I think there’s something wrong with the value 39.4 J/cm2, I think it’s higher: it cannot be equal to the first-order value mentioned at line 180.  Please check.

-Response9: This part is indeed wrong; the relevant section has been revised (line 152-167)

-Line 203: replace “Bezier” with “Bessel”.

Response10:"Bezier" have been deleted.

-Line 209: authors should mention that increased IR absorbance by fs-laser induced nanograting’s is well-known result from the literature (Appl.  Phys.  A 122, 211 (2016)).

- Response11: References have been added to the manuscript (line 181, reference 47-48)

Reviewer 3 Report

The paper calls: "Fabrication of Micro/nano Hybrid Structure for Broadband Antireflection by Multi-pulse Bessel Femtosecond Laser Direct writing" and concerned of method of reducing reflection of ZnS substrate by means of nanogratings formed by femtosecond radiation. The advantage of article are actual topic of research and long references list (38 ones).

Meanwhile it is not clear:

1. Why authors use verb "writing" instead of typical "inscription"?

2. String 146 1x105 (wrong format).

3. Authors explain effect of increasing transmission by forming microholes and nanogratings with period of 200-300 nm. Howerever, such relation between imperfections size (300 nm) and wavelength (5-12 microns) good fit for Relay scattering (relation less 1/10). What authors think about it?

4. Is it possible by such method of femtosecond holing produce structures wich be totally opac or transparent by means of changing period of holes?

Author Response

-Review1: Why authors use verb "writing" instead of typical "inscription"?

Response1: The words "Micromachining, fabrication, writing and inscription" are all commonly used, and the author just chose one of them.

-Review2: String 146 1x105 (wrong format).

Response2: The wrong format has been deleted(line131-132).

-Review3: Authors explain effect of increasing transmission by forming microholes and nanogratings with period of 200-300 nm.Howerever, such relation between imperfections size (300 nm) and wavelength (5-12 microns) good fit for Relay scattering (relation less 1/10). What authors think about it?

Response3: It is true that such the size of the nanostructure belongs to Rayleigh scattering (relation less 1/10). Because the nanostructure belongs to the periodic symmetric structure of one-dimensional grating, the light intensity is distributed symmetrically along the direction of incident light. This helps to compress the inner space of the grating structure, resulting in continuous collision and reflection of light on the inner wall of the structure [1], as shown in Fig.6 and Fig.7. This enhancement effect of local electromagnetic field can effectively capture the incident light and reduce the reflected light, thus achieving negative effect reduction.

It needs to be pointed out that light scattering collisions have various effects on different materials. For example, the transparent dielectric material of zinc sulfide, which almost does not absorb infrared light waves, can be considered that the absorption coefficient of its infrared band is zero. Therefore, the local enhancement effect of light wave collision will hardly cause the reflected light added to the transmitted light, enhancing transmission of zinc sulfide. For black silicon in the field of solar cells [2], silicon material absorbs light waves in the experimental band, so most of the reflected light, which reduced by the collision of light waves, will be absorbed by silicon material and not accumulated in the transmitted light.

-Review4: Is it possible by such method of femtosecond holing produce structures wich be totally opac or transparent by means of changing period of holes?

Response4: It is true that the transparent substrate can achieved by changing period of holes, which determined the spectral bandwidth of transmittance. However, it is not possible to achieve a completely opaque substrate only by changing the period, and the morphology of micro/nano structure also needs to be changed simultaneously (perhaps not the subwavelength microholes designed in the manuscript, which originate from the moth-eye structure of the bionic world). That is, a micro-nano structure that needs to be redesigned, which may be inspired by other animals or plants in the biomimetic world.

Round 2

Reviewer 2 Report

I’m happy with the revised version of the manuscript.

Authors have welcomed my suggestions and answered satisfactorily to my comments.

The English has improved significantly, authors  clarified the novelty of their work, and recalled correctly the concept of LIPSS when discussing about the laser-induced nano-structures.